# In Their Own Words: Reasoning Traces Tailored for Small Models Make Them Better Reasoners

## Abstract

Transferring reasoning capabilities from larger language models to smaller ones through supervised fine-tuning often fails counterintuitively, with performance degrading despite access to high-quality teacher demonstrations. We identify that this failure stems from distributional misalignment: reasoning traces from larger models contain tokens that are low probability under the student's distribution, exceeding the internal representation capacity of smaller architectures and creating learning barriers rather than helpful guidance. We propose Reverse Speculative Decoding (RSD), a mechanism for generating student-friendly reasoning traces in which the teacher model proposes candidate tokens but the student model determines acceptance based on its own probability distributions, filtering low probability tokens. When applied to Qwen3-0.6B, direct distillation of s1K-1.1 reasoning trace data degrades average performance across major reasoning benchmarks by 20.5%, while the same model trained on RSD-generated reasoning traces achieves meaningful improvements of 4.9%. Our analysis reveals that low probability tokens constitute the critical bottleneck in reasoning ability transfer. However, cross-model experiments demonstrate that RSD traces are model-specific rather than universally applicable, indicating that distributional alignment must be tailored for each student architecture's unique internal representation. Code and datasets are available at `https://anonymous.4open.science/r/rsd`.

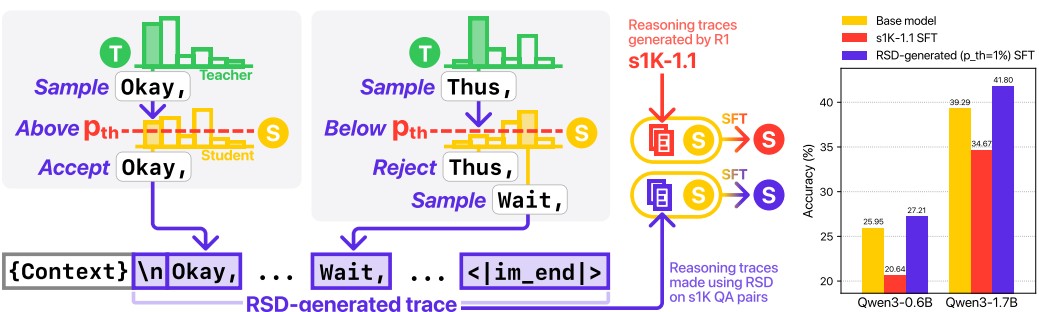

Figure 1: **Conceptual overview and empirical validation of Reverse Speculative Decoding (RSD).** Left: Reasoning trace generation process where RSD produces student-friendly reasoning traces in which the teacher proposes candidate tokens, while the student accepts only those with high probability under its own distribution. Right: Average accuracy on major reasoning benchmarks (AIME24, AIME25, GPQA Diamond, and MATH500) for (i) the base student model, (ii) a student trained on pre-existing high-quality reasoning traces (s1K-1.1), and (iii) a student trained on the reasoning traces it helped generate through the RSD process shown on the left.

## 1 Introduction

Recent advances in reasoning-focused language models have emerged through the strategic combination of reinforcement learning (RL) and supervised fine-tuning (SFT) (DeepSeek-AI et al., 2025). These two methods play distinct yet complementary roles in developing sophisticated reasoning.

While RL excels at eliciting reasoning capacities by encouraging the model to explore and reflect, SFT is paramount in instilling reasoning abilities through direct exposure to expert demonstrations.

When model capacity is limited, SFT assumes a more prominent role. Although RL can still contribute to reasoning ability, it often requires far more data and compute to reach comparable levels, with diminishing returns as model size shrinks. In contrast, SFT enables compact architectures to efficiently inherit problem-solving strategies from more capable teachers. Experimental results on 32B models suggest that small models trained with RL on complex reasoning tasks often lag behind peers distilled from high-performing teachers, even when granted greater training resources (DeepSeek-AI et al., 2025). Consequently, leveraging intricate reasoning traces from capable, large models to train smaller models has become a dominant strategy for effective reasoning transfer.

However, empirical evidence reveals significant limitations in this transfer approach when working with even smaller models with just a few billion parameters. While approaches utilizing small collections of carefully curated reasoning traces, specifically s1K (Muennighoff et al., 2025) and LIMO (Ye et al., 2025), have demonstrated success with 32B models, these same datasets reveal a starkly different outcome when applied to substantially smaller 3.8B architectures (Xu et al., 2025a). When these compact models attempt to learn from high-quality reasoning traces distilled from larger teachers, direct distillation can significantly degrade performance, creating a phenomenon where models paradoxically deteriorate despite access to superior training data. This counterintuitive regression suggests that the reasoning behaviors naturally emerging in large models may prove ill-suited for direct imitation by substantially smaller counterparts, where the elaborate reasoning patterns and long logical dependencies can overwhelm compact architectures, causing capability regression.

We posit that the fundamental challenge lies in the leap in perceived complexity across consecutive reasoning steps that student models encounter. In language modeling, this disparity manifests at the token level. When the teacher's next token falls in a region of very low probability under the student's distribution, it may signal a reasoning pattern that exceeds what the student's current internal representation can process. Effective transfer requires reshaping the stride of reasoning steps so that the rise in difficulty remains locally smooth—keeping the cognitive load between steps equigranular from the student's perspective. Rather than compelling a small model to recite a teacher's reasoning verbatim, we advocate for creating traces that preserve correctness while ensuring each reasoning transition remains tractable within the student's processing range.

In this work, we propose Reverse Speculative Decoding (RSD), a novel algorithm for generating such student-friendly traces, and a training recipe to effectively transfer reasoning ability to smaller student models. As illustrated in Figure 1, in RSD, the teacher proposes a token, but the student decides whether to accept it based on its own probability distribution; if the token has the probability below a certain threshold, it is deemed improbable by the student, and the generation falls back to the student's own prediction. This inverted teacher–student dynamic ensures that teacher guidance is injected only where the student is ready to follow, promoting distributional alignment and thus producing reasoning steps aligned with the student's representational capacity.

We demonstrate the effectiveness of RSD through comprehensive experiments across major reasoning benchmarks. Our findings reveal that while direct SFT on raw teacher traces leads to performance degradation, RSD-generated traces consistently improve reasoning capability. Our experiments show that the optimal configuration uses the probability threshold of $p_{\text{th}}=1\%$ with a temperature $T=0.7$, striking the balance between filtering low probability tokens and preserving teacher guidance. These findings underscore that such low-probability tokens represent the critical bottleneck to effective reasoning transfer, validating our threshold-based filtering approach.

## 2 RELATED WORK

**Reasoning Trace Rewrite**  As supervised fine-tuning on reasoning traces became prevalent, the quality of training data emerged as a critical factor for performance improvement. This recognition sparked extensive research into generating superior reasoning traces through various conditioning and rewriting strategies. Some approaches focused on efficiency, generating shorter yet equally effective reasoning chains through summarization (Kang et al., 2025) or self-training with best-of-$n$ selection (Munkhbat et al., 2025). Others pursued targeted improvements, employing difficulty-aware prompting during trace generation (Wu et al., 2025) or conditioning on behavior handbooks

or reasoning templates that provide task-specific reasoning guidelines (Didolkar et al., 2025). More sophisticated approaches adopted MCTS-inspired generation strategies to eliminate redundant reasoning steps and explore alternative reasoning paths (Lu et al., 2025).

Despite these advancements, we believe there is a largely underexplored angle in this space: generating easier reasoning traces where each logical leap is narrower and more accessible to smaller models. Our approach focuses on ensuring that reasoning demonstrations align with what small models can readily follow and learn from, in order to transform them into better reasoners.

**Teacher-Student Coordination**   Teacher-student coordination mechanisms have been explored across both training and inference phases. At test-time, speculative decoding (Leviathan et al., 2023) accelerates inference by having smaller models propose token candidates for verification by larger models. Step-level coordination approaches include methods where larger models intervene during detected reasoning difficulty through structural cues (Yang et al., 2025b), or where smaller models learn to emit special tokens requesting help from larger models (Akhauri et al., 2025). These approaches leverage the observation that not all generation steps need equal computational resources.

The principles underlying these test-time coordination strategies have also been adapted for training data generation. To reduce distributional mismatch between training and inference, Speculative Knowledge Distillation (SKD) (Xu et al., 2025b) employs student-proposed, teacher-approved sampling. While this creates higher-quality training contexts, the teacher-centric approval can still force students along unnatural trajectories. Closely related to our goal, Reinforcement-Learned Teachers (RLT) (Cetin et al., 2025) explicitly align distributions by training the teacher with a KL-divergence constraint, yet this requires expensive retraining. In contrast, RSD operates purely as a data synthesis mechanism, utilizing the teacher as-is to generate aligned training data without the heavy cost of modifying teacher weights. Our approach prioritizes distributional alignment through teacher-proposed, student-approved generation—hence the name, Reverse Speculative Decoding.

## 3   METHOD

### 3.1   GENERATING STUDENT-FRIENDLY TRACES WITH RSD

The core principle of RSD is that effective reasoning ability transfer requires managing the surprisal experienced by student models during learning. Algorithm 1 operationalizes this principle through a teacher-proposed, student-approved generation mechanism. At each decoding step, we obtain probability distributions from both the teacher model $P_t$ and student model $P_s$, then sample a candidate token $y_i \sim P_t$ and evaluate its likelihood under the student model $P_s(y_i)$. If $P_s(y_i) \geq p_{\text{th}}$, we accept the teacher's proposal; otherwise, we fall back to sampling directly from the student distribution $y_i \sim P_s$.

---

**Algorithm 1** Reverse Speculative Decoding

**Require:** Teacher LLM $M_t$, Student LLM $M_s$, Prompt $x$, Probability threshold $p_{\text{th}}$, Decoding length $\alpha$
1:  context $\leftarrow x$
2:  **for** $i = 1$ to $\alpha$ **do**
3:      $P_t \leftarrow M_t(\cdot|\text{context})$
4:      $P_s \leftarrow M_s(\cdot|\text{context})$
5:      $y \sim P_t$
6:      **if** $P_s(y) < p_{\text{th}}$ **then**
7:          $y \sim P_s$
8:      **end if**
9:      context $\leftarrow$ context $+ y$
10:     Break if $y = $ EOS
11: **end for**
12: **return** context

---

This selective acceptance mechanism ensures distributional alignment throughout the generated trace. We can conceptualize the cognitive load at each step as the surprisal $\ell_i = -\log P_s(y_i)$, with the threshold load being $-\log p_{\text{th}}$. By filtering tokens that exceed this threshold, RSD effectively smooths surprisal spikes that would otherwise create learning obstacles. Each accepted teacher token represents a reasoning step within the student's internal representation, while rejected tokens signal transitions that would create excessive uncertainty.

To ensure both correctness and student-friendliness in the generated traces, we employ rejection sampling, generating multiple candidate traces per problem and selecting a correct one for training (Yuan et al., 2024). This approach produces reasoning demonstrations that are both distributionally aligned and semantically sound.

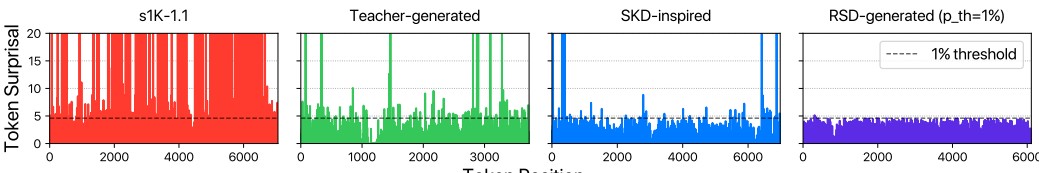

Figure 2: **Token-level surprisal progression across different trace generation methods.** Comparison of token surprisal patterns for a student model across traces generated by different methods. RSD's effectiveness in eliminating problematic high-surprisal spikes that create learning barriers for student models is demonstrated.

## 3.2 Quantifying Distributional Alignment

To analyze the distributional characteristics of reasoning traces, we employ several complementary metrics that capture different aspects of the student model's uncertainty:

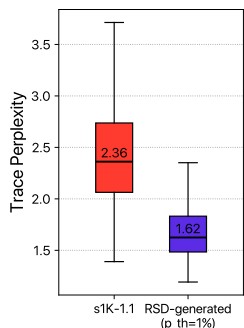

**Surprisal and Entropy** Following Shannon's work on information theory (Shannon, 1948), we compute the surprisal of each token $y_i$ in a trace under the student model as $s_i = -\log P_s(y_i|y_{<i})$. The entropy of the student's distribution at each step is given by $H = -\sum P_s(y_i|y_{<i})\log P_s(y_i|y_{<i})$. High surprisal indicates tokens that fall in low-probability regions of the student's distribution, representing potential learning obstacles. These information-theoretic measures both capture regions where the student model exhibits substantial uncertainty about the next step. Figure 2 illustrates how RSD effectively eliminates problematic high-surprisal spikes compared to other trace generation methods, demonstrating the mechanism's ability to smooth token-level surprisal progression throughout reasoning traces.

Figure 3: **Trace-level perplexity distributions.** RSD-generated traces cluster at lower perplexity values with reduced variance.

**Perplexity** At the trace level, we compute perplexity as $\text{PPL} = \exp(\frac{1}{N}\sum_{i=1}^{N} s_i)$, where $N$ is the trace length. This provides a measure of how well-aligned an entire reasoning trace is with the student's distribution. Lower perplexity indicates traces are more natural from the student's perspective. In Figure 3, RSD traces consistently cluster at lower perplexity values with reduced variance compared to baseline methods, providing an evidence of improved distributional alignment.

**Sub-threshold Token Ratio** We track the proportion of tokens with probability below 1% under the student model. Our empirical findings reveal this metric as the strongest predictor of learning failure, with traces containing many sub-1% tokens consistently degrading student performance.

## 3.3 Maximizing Learning Signal with a Hybrid Training Approach

RSD approaches reasoning transfer through a trace rewriting process, reconstructing teacher demonstrations through student distributional constraints. However, this constrains the teacher's problem-solving reach—even with 16 rejection samples, RSD cannot solve all problems. This limitation actually validates our approach—if RSD solved everything under distributional constraints, the problems would lack sufficient complexity. Rather than discarding unsolved problems, we employ a dual-component methodology that maximizes the utilization of available training signal:

**Primary RSD Training** For problems where RSD generates correct solutions, we train on complete traces using standard SFT, ensuring both logical correctness and distributional alignment.

**UPFT for Unsolved Problems** For problems where RSD fails to generate a correct solution, we employ a partial trace training strategy to salvage the valuable reasoning patterns present in the initial steps. Inspired by the Unsupervised Prefix Fine-Tuning (UPFT) methodology (Ji et al., 2025), we extract the first 128 tokens from these unsuccessful traces. This approach ensures no training instances are wasted, allowing the student model to learn how to recognize problem patterns and formulate initial approaches even from examples that don't reach a correct final answer.

## 4 EXPERIMENTS

### 4.1 SETUP

**Teacher-Student Model Pair**  RSD requires tokenizer compatibility between teacher and student models, as each teacher-proposed token must be evaluated under the student's probability distribution. For our main experiments, we employ s1.1-7B (Muennighoff et al., 2025), a Qwen2.5-7B variant fine-tuned on s1K-1.1, as our teacher model and Qwen3-0.6B (Yang et al., 2025a) as our student model. These models share a tokenizer, enabling the token-level probability evaluations essential to the RSD mechanism. Details on tokenizer compatibility can be found in Appendix A.

**Baselines**  We select the s1K dataset (Muennighoff et al., 2025) containing 1,000 challenging problems spanning mathematics, science, and logic that demand sophisticated reasoning rather than simple pattern recognition. We use s1K-1.1—traces generated by DeepSeek-R1 on the s1K questions—as our primary baseline, though our method generalizes to any dataset providing meaningful learning signals. Using s1K's question-answer pairs, we generate RSD traces through rejection sampling with temperature $T$=0.7 and probability thresholds $p_{th} \in \{10\%, 3\%, 1\%, 0.3\%\}$.

To isolate RSD's impact, we also compare against: (1) teacher-generated traces using our 7B teacher model to assess whether smaller teacher capacity drives RSD's effectiveness, (2) student-generated self-distillation traces to evaluate training on student's own outputs, and (3) SKD-inspired generation implementing student-proposed, teacher-approved dynamics with 1% probability threshold—more restrictive than standard SKD's top-$k$ sampling. All trace generation uses an 8k token context limit.

**Model Training**  Following the s1 training recipe (Muennighoff et al., 2025), we use batch size 16, bfloat16 precision, learning rate $1 \times 10^{-5}$ with 5% linear warmup followed by cosine decay, AdamW optimizer ($\beta_1$=0.9, $\beta_2$=0.95, weight decay $10^{-4}$). We train for 15 epochs.

### 4.2 DISTRIBUTIONAL ALIGNMENT DRIVES RSD EFFECTIVENESS AND DATA EFFICIENCY

Table 1: **Impact of different distillation methods and RSD probability thresholds on the reasoning performance of the Qwen3-0.6B model.** Direct distillation, where the student is fine-tuned on unaltered reasoning traces from teacher models (s1K-1.1, Teacher-generated), consistently degrades performance. In contrast, RSD-generated traces yield improvements, with a probability threshold of 1% achieving the best average performance. Evaluation details are available in Appendix B. Best results are in **bold**, second best are underlined.

| Models | AIME24 | AIME25 | GPQA Diamond | MATH500 | Average |
|---|---|---|---|---|---|
| Qwen3-0.6B | 2.71 | 10.94 | 24.75 | 65.40 | 25.95 |
| + s1K-1.1 | 1.93 | 9.53 | 12.88 | 58.20 | 20.64 |
| + Teacher-generated | 1.35 | 8.91 | 12.31 | 58.80 | 20.34 |
| + Self-distill | 2.66 | 10.78 | 21.97 | **67.80** | 25.80 |
| + SKD-inspired | 2.40 | 11.56 | 4.17 | 65.40 | 20.88 |
| + RSD-generated ($p_{\text{th}}$=10%) | **3.33** | 11.25 | 24.87 | 66.20 | 26.41 |
| + RSD-generated ($p_{\text{th}}$=3%) | 2.97 | 11.56 | 24.24 | 66.80 | 26.39 |
| + RSD-generated ($p_{\text{th}}$=1%) | 3.28 | **12.60** | **26.77** | 66.20 | **27.21** |
| + RSD-generated ($p_{\text{th}}$=0.3%) | 1.41 | 9.53 | 23.04 | 63.80 | 24.45 |

**RSD with 1% probability threshold achieves optimal performance by balancing token filtering with meaningful teacher guidance.**  As shown in Table 1, the 1% threshold configuration demonstrates the most significant improvements across all benchmarks for our 0.6B student model, while higher thresholds of 10% and 3% show less consistent gains, and the restrictive 0.3% threshold causes substantial degradation. This performance pattern directly correlates with the sub-1% probability token ratios presented in Table 2, where the 0.3% threshold fails to adequately filter problematic tokens (2.02% sub-1% tokens), while optimal configurations maintain extremely low ratios (0.04–0.09%). The sub-1% tokens metric represents the proportion of all tokens in each dataset that have probability below 1% under the student model's distribution, serving as a strong predictor of learning failure in compact architectures.

Table 2: **Dataset characteristics for different trace generation methods showing problem coverage, fallback rates, and sub-1% probability token proportions.** The s1K-1.1 traces contain a high proportion of sub-1% tokens, which correlates with poor training outcomes, whereas all RSD variants drastically reduce this proportion.

| Datasets | Correctly solved | Fallback rate (%) | Sub-1% tokens (%) |
|---|---|---|---|
| s1K-1.1 | 1000 | Not Applicable | 6.70 |
| Teacher-generated | 234/1000 | Not Applicable | 2.98 |
| Self-distill | 122/234 | Not Applicable | 0.00 |
| SKD-inspired | 184/234 | 0.68 | 0.72 |
| RSD-generated ($p_{th}$=10%) | 161/234 | 2.71 | 0.06 |
| RSD-generated ($p_{th}$=3%) | 171/234 | 1.28 | 0.04 |
| RSD-generated ($p_{th}$=1%) | 180/234 | 0.64 | 0.09 |
| RSD-generated ($p_{th}$=0.3%) | 177/234 | 0.35 | 2.02 |

The number of correctly solved problems during trace generation provides some indication of RSD effectiveness. Correctly solved metric in Table 2 indicates how many problems each method successfully generates correct solutions for during the trace generation process. Since RSD requires both teacher and student model coordination, we first let the teacher model solve the 1,000 s1K problems, successfully obtaining solutions for 234 problems, which explains the /234 notation for methods that depend on teacher-generated solutions. Student-generated self-distill traces operate independently of teacher performance, but we applied the same constraint based on our assumption that the student model can only reasonably solve problems that the teacher has already solved. Among these problems, RSD with 1% threshold generated correct solutions for 180 problems—the highest among all RSD configurations. However, SKD-inspired generation solved 184 problems while still underperforming RSD during model training, demonstrating that correctness-preserving generation comes at the cost of higher sub-1% token ratios (0.72%), which creates learning barriers for compact architectures.

Traces from both large and smaller teacher models create equal distributional misalignment when training compact students. Both s1K-1.1 traces (generated by 671B DeepSeek-R1) and our teacher-generated traces (7B model) exhibit similar poor performance when training the 0.6B student. This similarity suggests that teacher model capacity does not reduce the degree of distributional misalignment—traces from both large and smaller teachers create equal learning barriers for compact students despite their substantial capacity difference. SKD-inspired generation also demonstrates low performance, primarily due to extremely poor GPQA Diamond scores where models frequently failed to produce answers within the token budget. Even excluding these failures, SKD-inspired still underperforms RSD 1% across all other benchmarks.

Fallback rates demonstrate variation across RSD probability thresholds. Fallback rates—the proportion of tokens where teacher proposals fall below the probability threshold, causing generation to revert to the student model—remain consistently low across all RSD configurations (0.35% to 2.71%). More restrictive thresholds result in lower fallback rates, with the 0.3% configuration showing the lowest rate (0.35%) and the 10% configuration showing the highest (2.71%).

The non-zero sub-1% token ratios in RSD traces likely occur when teacher influence introduces subtle perturbation that nudges generation away from the student's natural distribution, causing the student to select low-probability tokens during fallback generation. This contrasts with self-distill's near-zero sub-1% ratio, which reflects purely student-native generation without external guidance.

**RSD achieves meaningful improvements for small models using remarkably few examples compared to existing approaches.** While methods like Phi-4-Mini-Reasoning (Xu et al., 2025a) require extensive training from mid-training onwards with massive datasets to develop reasoning capabilities in compact models, RSD demonstrates that targeted filtering can produce improvements using only 1,000 carefully curated examples. This efficiency becomes even more striking considering that among these examples, only 180 are complete reasoning traces, while the remainder consists of 128 token prefixes. Such efficiency emerges from RSD's targeted approach: rather than overwhelming compact models with vast quantities of reasoning data, the method precisely identifies and removes the specific elements that create learning barriers.

**The probability threshold mechanism is instrumental in addressing the fundamental challenge of reasoning transfer to compact architectures.** When a teacher's token has probability below

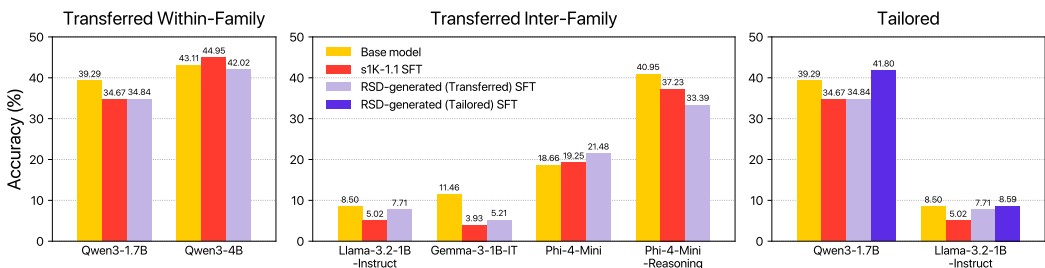

Figure 4: **Cross-model experiments demonstrate the model-specific nature of RSD-generated traces.** Reasoning traces generated using one student model (Qwen3-0.6B) fail to transfer benefits when applied to other models, both within the same model family (Left) and across different families (Center). When the RSD process is tailored to each student model, it produces performance gains (Right). Average accuracy of major reasoning benchmarks are shown. Detailed evaluation results are available in Appendix D.

1% under the student distribution, it may represent an abrupt reasoning pivot, a shift in analytical perspective, or an alternative exploratory direction that exceeds the student's internal representation capacity. In our findings, s1K-1.1 traces contain 6.70% sub-1% probability tokens and degrade student performance by 20.5% (from 25.95% to 20.64% average accuracy), while RSD with 1% threshold reduces sub-1% tokens to just 0.09% and achieves meaningful improvements of 4.9% (to 27.21% average accuracy). By systematically filtering these high-surprisal tokens while preserving solution correctness through rejection sampling, RSD creates traces where each reasoning transition remains within the student's processing range.

## 5 ANALYSIS

### 5.1 CROSS-MODEL TRANSFER OF RSD-GENERATED TRACES

To investigate whether RSD traces represent universally accessible reasoning patterns, we test whether traces generated using Qwen3-0.6B as the student can benefit other student models during training. This evaluation encompasses two dimensions of architectural variation, thus providing an empirical test of the cognitive load hypothesis and offering insight into whether RSD traces genuinely ease the reasoning burden for compact learners.

**Within-Family Transfer** We evaluate how RSD traces transfer across different scales within the Qwen3 family, testing on 1.7B and 4B parameter variants. This tests whether the distributional alignment achieved for a 0.6B model provides benefits for large models in the same model family.

**Inter-Family Transfer** We apply RSD traces originally generated with Qwen3-0.6B as the student to fundamentally different architectures: Llama-3.2-1B-Instruct (AI at Meta, 2024), Gemma-3-1B-IT (Team et al., 2025), Phi-4-Mini (Microsoft et al., 2025), and Phi-4-Mini-Reasoning (Xu et al., 2025a). The first three models are designed for general-purpose tasks and not specifically for reasoning tasks, while Phi-4-Mini-Reasoning is a specialized reasoning model that natively uses the thinking delimiters. Through SFT, the non-reasoning models learn to adopt this structured reasoning approach with the thinking delimiters, effectively transforming them into reasoning-capable models.

Figure 4 delivers a crucial finding: RSD traces are model-specific rather than universally beneficial. While traces generated using Qwen3-0.6B improve the original student model, they consistently fail to transfer benefits when applied to other models. The failure extends across both inter-family and within-family evaluations, revealing that distributional alignment is an inherently model-specific phenomenon dependent on the characteristics of each model's learned probability distribution. Additionally, we observe that for the Qwen3-4B model, direct distillation yields improvements unlike the 0.6B model. The 4B model has sufficient internal representation capacity to process the logic from the s1K-1.1 teacher directly. Consequently, RSD specifically targets students with limited representational capacity, particularly those with fewer than 4 billion parameters, where direct distillation fails due to capacity constraints.

## 5.2 Model-Dependent RSD Performance

Given the model-specific nature of RSD traces demonstrated in Figure 4, we investigate whether the RSD method itself proves effective when applied to different student architectures. Due to the tokenizer compatibility constraint inherent to the RSD mechanism, the range of different student models we can experiment with is limited. We choose Llama-3.2-1B-Instruct as our student model paired with DeepSeek-R1-Distill-Llama-8B as the teacher. This combination of a larger reasoning-focused teacher model with a compatible student model represents a relatively unique pairing in the current model landscape. We also experiment with Qwen3-1.7B as our student model with the s1.1-7B teacher, leveraging the fact that models within the same Qwen3 family naturally share vocabulary with our primary Qwen3-0.6B student model. As seen from Figure 4, Qwen3-1.7B demonstrates notable improvement when trained on its own RSD-generated traces, while Llama-3.2-1B-Instruct exhibits minimal improvements despite identical RSD methodology.

This contrasting behavior reveals that RSD effectiveness depends critically on architectural characteristics. As detailed in Appendix F, Llama-3.2-1B-Instruct exhibits inherently terse reasoning traces approximately four times shorter than Qwen3 counterparts, reflecting different linguistic preferences that influence the generation of reasoning demonstrations. This concise expression style reflects the model's design for general-purpose tasks and training data that predates DeepSeek-R1, lacking exposure to the extended inner monologue patterns now characteristic of recent reasoning-focused models. These findings highlight an important design principle of the RSD mechanism: it operates by working within a student's existing distributional preferences rather than attempting to impose fundamentally different linguistic behaviors. The student-centric approach of RSD naturally preserves each model's inherent reasoning style, allowing the method to enhance existing patterns while respecting the architectural boundaries established during pre-training.

## 5.3 Implications for the Universal Cognitive Load Hypothesis

One hypothesis for reasoning ability transfer posits that cognitive load—the mental effort required to process conceptual leaps and logical transitions between consecutive reasoning steps—represents a universal limiting factor that affects all reasoning agents, human learners and language models alike. Under this framework, methods like RSD can be expected to produce universally beneficial reasoning demonstrations by reducing cognitive load through more manageable reasoning progressions. However, the cross-model transfer results in Figure 4 challenge this notion. They reveal that distributional alignment is an inherently model-specific phenomenon where traces tailored for one model's internal representation do not transfer to another's, even within the same model family. The failure of these traces to transfer indicates that each model develops unique internal representations during pre-training, where effectiveness depends on the precise characteristics of each model's learned probability distribution rather than abstract cognitive demands. What constitutes a natural reasoning step for one model may represent an inexplicable leap for another, even when both models operate at similar parameter scales, suggesting that reasoning transfer barriers are fundamentally architectural rather than universally cognitive.

## 5.4 Multi-Step RSD Training

We explore iterative RSD application through a multi-step training approach, using Qwen3-0.6B with the optimal 1% probability threshold for three complete cycles, with each cycle consisting of 5 training epochs and the trained model serving as the new student for subsequent RSD trace generation. Complete results are available in Appendix E. Performance degraded substantially due to compounding effects that make iterative alignment inherently problematic. Since RSD generates traces aligned to a student's current distributional characteristics, repeated application reinforces increasingly specialized patterns rather than broadening capabilities, leading to progressive overfitting. The degradation during iterative training should be viewed in the context of the broader, unsolved research challenge of Model Collapse (Shumailov et al., 2024). When a model is recursively trained on its own output distribution, even with teacher guidance, the distribution tends to narrow, reducing diversity and capability. We believe this represents a fundamental limitation of consecutive alignment strategies rather than the specific RSD mechanism.

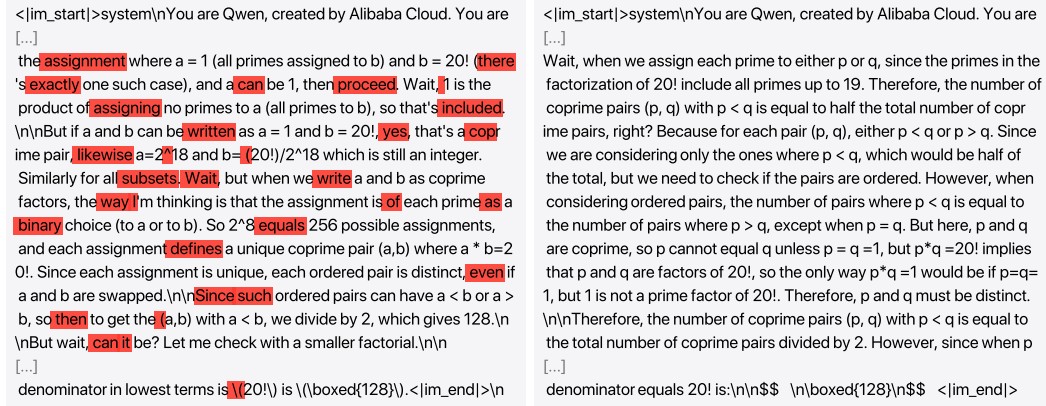

Figure 5: **Comparison of trace excerpts from the same question demonstrates RSD's distributional alignment.** The s1K-1.1 trace excerpt (left) contains numerous sub-1% probability tokens (red highlights) while the corresponding RSD trace excerpt (right) exhibits smooth probability transitions.

### 5.5 CHARACTERISTICS OF HIGH-SURPRISAL BOTTLENECKS

As seen from Figure 5 and Appendix G, analysis reveals that high-surprisal tokens often correspond to critical junctures in reasoning such as logical connectors that fork the reasoning path. Recent works (Wang et al., 2025b;a) have identified similar patterns, showing that tokens with high entropy frequently mark critical decision points where multiple possible continuations exist. In the context of reasoning ability transfer, these branching points become particularly problematic, as while a large model can navigate complex logical forks based on its extensive internal representation, smaller models lack the capacity to represent all possible branches simultaneously.

### 5.6 QUALITATIVE ANALYSIS OF RSD TRACES AND LOGICAL TRADE-OFFS

To investigate whether RSD's filtering impacts the semantic quality of reasoning, we conducted a pairwise comparison using GPT-5-nano as a judge. We sampled trace pairs (s1K-1.1 versus RSD-generated ($p_{\text{th}}$=1%)) and asked the evaluator to compare them on logical correctness and tractability.

Table 3: **Qualitative evaluation of trace quality via GPT-5-nano judge.** RSD traces exhibit a slight degradation in logical correctness and tractability compared to the original teacher traces.

| Metric | s1K-1.1 | RSD-generated ($p_{\text{th}}$=1%) |
|---|---|---|
| Logical correctness | 53.89% | 46.11% |
| Logical tractability | 51.67% | 48.33% |

The results in Table 3 show that RSD traces exhibit a slight degradation in both logical correctness and tractability compared to the original teacher traces. When the student rejects a teacher token and samples from its own distribution, it risks choosing a logically incorrect path. This is evidenced by the drop in the number of correctly solved problems during generation, as shown in Table 2.

However, this trade-off is central to the RSD mechanism. Our experiments show that training on these fewer but aligned traces leads to a better final model. The student learns more effectively from slightly imperfect but understandable traces than from perfect traces it cannot comprehend.

### 5.7 GENERALIZATION BEYOND MATHEMATICAL AND SCIENTIFIC REASONING

In order to demonstrate the efficacy of our method beyond mathematical and scientific reasoning, we conducted additional experiments on MMLU-Pro (Wang et al., 2024), specifically selecting categories requiring logical deduction: Law, Economics, Health, Psychology, Business, and Philosophy.

The trend observed in math and science benchmarks holds for general domain reasoning as detailed in Appendix H. Direct distillation of s1K-1.1 traces degrades performance from the base model accuracy of 30.93% to 30.54%, whereas training on RSD-generated traces with a 1% probability threshold improves accuracy to 31.39%. This confirms that the distributional alignment provided by RSD remains beneficial across diverse reasoning domains.

## 5.8 Robustness Across Datasets

To validate that RSD's effectiveness is not an artifact of the s1K dataset, we conducted a new experiment using OpenThoughts3-1.2M. We processed the dataset by selecting only math and science domains, filtering out truncated traces that exceeded the context limit, and deduplicating to ensure unique question-answer pairs. From the resulting pool of approximately 35k valid traces, we randomly sampled 1,000 examples to ensure a fair, size-controlled comparison with our s1K baseline.

Table 4: **Robustness validation on OpenThoughts3-1.2M (1k subset).** Direct distillation of OpenThoughts3-1.2M traces causes severe degradation due to distributional misalignment. RSD-generated traces on the question-answer pairs of OpenThoughts3 mitigates this failure and achieves the highest performance on the most challenging benchmark (AIME24).

| Models | AIME24 | AIME25 | GPQA Diamond | MATH500 | Average |
|---|---|---|---|---|---|
| Qwen3-0.6B | 2.71 | 10.94 | 24.75 | 65.40 | 25.95 |
| + OpenThoughts3 | 1.82 | 6.35 | 22.41 | 54.00 | 21.15 |
| + RSD-generated (OpenThoughts3, RSD $p_{th}$=1%) | 4.01 | 9.58 | 22.92 | 64.40 | 25.23 |

The results in Table 4 reinforce our core hypothesis: Direct distillation causes severe degradation, even when utilizing OpenThoughts3—a dataset distinct from s1K and released after the Qwen3 family. This confirms that distributional misalignment remains a critical barrier independent of dataset specificities. While RSD on this specific sample did not exceed the base model's average score, it successfully mitigated the degradation caused by direct distillation. Notably, on AIME24—the most challenging benchmark requiring the deepest reasoning—RSD achieved the highest performance (4.01), outperforming both the base model (2.71) and direct distillation (1.82). This suggests that RSD is particularly effective at preserving the complex reasoning required for hard problems.

## 5.9 Isolating Distributional Alignment from Computational Investment

While computational costs involved in trace generation represent secondary concerns in reasoning trace research, one might attribute RSD's effectiveness to increased computational investment rather than distributional alignment. To test this hypothesis, we provide the evaluation results from student-generated self-distill rejection sampling with 203 attempts instead of 16 to match RSD's computational budget. Despite solving more problems than RSD 1% (189/234 versus 180/234), model performance remained unchanged and continued to underperform the base model. RSD remains the only method that consistently improves upon baseline performance under compute-equivalent conditions. Details of this compute equivalence analysis can be found in Appendix C.

## 6 Conclusion

We introduced Reverse Speculative Decoding (RSD) to address distributional misalignment in reasoning ability transfer. By filtering high-surprisal tokens that exceed student models' internal representation capacity, RSD transforms teacher traces into student-friendly demonstrations while preserving logical correctness. Our findings reveal that effective reasoning transfer hinges on managing token-level surprisal, with sub-1% probability tokens serving as reliable indicators of representational incompatibility. We also identified the model-specific nature of RSD where these benefits requires tailored trace generation for each model. We believe our work opens up new avenues for reasoning ability transfer research, bringing distributional alignment to the forefront as a critical consideration for effective distillation in compact architectures.

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

## A  Tokenizer Compatibility Details

The RSD mechanism requires precise token-level probability evaluation, making tokenizer compatibility between teacher and student models essential. Without compatible tokenizers, token-level comparison becomes infeasible, as vocabulary from one model may not exist in, or is different from, another. Converting tokens back to text for comparison creates problems because BPE tokenizers often split rare symbols (especially mathematical ones) across multiple sub-tokens, and this reconstruction process can fail since sub-tokens represent incomplete character fragments rather than standalone symbols.

Even though our teacher model s1.1-7B and student model Qwen3-0.6B are from the same model family, incompatible token IDs exist between them due to different training procedures. We handle these vocabulary discrepancies through several technical adjustments:

**Vocabulary Suppression:** We suppress 128 extra entries present only in the teacher model's vocabulary during generation to ensure all teacher-proposed tokens can be evaluated by the student model.

**Separate Context Management:** For tokens unique to the student vocabulary (specifically IDs 151665–151668, which include thinking delimiters like `<think>` and `</think>`), we maintain separate contexts for teacher and student models during generation. This ensures that both models can process the reasoning traces in their native token formats while enabling probability evaluation.

**Token Mapping Example:** Token ID $151668$ corresponds to `</think>` in the student model but maps to the sequence $(522, 26865, 29) = (\texttt{</}, \texttt{think}, \texttt{>})$ in the teacher's tokenizer. During RSD generation, we preserve the native token format in the student's context while using the mapped representation in the teacher's context, ensuring both models can process the same semantic content.

These technical considerations highlight why tokenizer compatibility represents a practical constraint for RSD implementation, limiting the range of teacher-student model pairs that can be effectively used with this approach.

## B  Evaluation Details

We assess performance on four challenging benchmarks: AIME24 and AIME25 (competition mathematics), GPQA Diamond (Rein et al., 2024) (graduate-level science), and MATH500 (Hendrycks et al., 2021) (diverse mathematical reasoning). We report avg@64 for AIMEs, avg@8 for GPQA Diamond, and pass@1 for MATH500. Context limits are 8k tokens for all benchmarks but 16k for GPQA Diamond to accommodate extended reasoning processes.

For GPQA Diamond, a multiple-choice dataset with deliberately crafted distractors, we implement a special handling: if models haven't produced a definitive answer by 15k tokens, we forcibly insert `</think>` to terminate the thinking phase and encourage answer generation. This prevents models from reasoning indefinitely and ensures fair comparison against the 25% random baseline.

## C  Compute-Equivalent Setting

To ensure RSD's effectiveness stems from distributional alignment rather than sheer computational investment, we conducted a compute-equivalent comparison. Our best-performing RSD configuration (1% threshold), which uses a 7B teacher and a 0.6B student with 16 rejection samples, was benchmarked against student-only self-distillation. To match the computational budget, we allocated the self-distillation method an increased number of attempts, calculated as $((7/0.6)+1) \times 16 \approx 203$ samples. The results in Table 5 show that despite this significantly larger budget and solving more problems during trace generation (189/234 versus 180/234), the compute-equivalent self-distillation method failed to improve performance over its baseline and continued to underperform the base model. This isolates RSD's benefits to its alignment mechanism, confirming that trace quality is more critical than trace quantity or the computational cost of generation.

Table 5: **Compute-equivalent comparison between RSD and student-generated rejection sampling.** Despite the increased budget, self-distillation fails to improve over the base model, demonstrating that RSD's effectiveness stems from its alignment mechanism, not merely from increased computational investment.

| Models | AIME24 | AIME25 | GPQA Diamond | MATH500 | Average |
|---|---|---|---|---|---|
| Qwen3-0.6B | 2.71 | 10.94 | 24.75 | 65.40 | 25.95 |
| + Self-distill (16 rejection sampling attempts) | 2.66 | 10.78 | 21.97 | 67.80 | 25.80 |
| + Self-distill (203 rejection sampling attempts) | 2.55 | 11.09 | 23.30 | 66.80 | 25.94 |
| + RSD-generated ($p_{\text{th}}$=1%) | 3.28 | 12.60 | 26.77 | 66.20 | 27.21 |

## D  Cross Model Evaluation Results

We conducted cross-model evaluations, detailed in Table 6, to test if RSD-generated traces are universally beneficial. The results show that traces are highly model-specific; those generated for one student model failed to improve others and often degraded performance. However, tailoring the RSD process to a new student model yielded significant gains. This demonstrates that effective reasoning transfer requires distributional alignment to be specifically calibrated for each student architecture.

## E  Multi-Step RSD Training Results

To investigate if RSD's benefits could be compounded, we tested an iterative multi-step training approach. The experiment consisted of three complete cycles using the Qwen3-0.6B model, where the trained model from each cycle served as the new student for the next round of trace generation. Each cycle was trained for 5 epochs, maintaining the optimal probability threshold of $p_{\text{th}} = 1\%$. As detailed in Table 7, the results show that this iterative process substantially degrades performance, falling below both the single-step RSD model and the original baseline. This suggests that repeated re-alignment creates a detrimental feedback loop, leading to issues like compounding distributional drift and overfitting to narrow reasoning patterns, which prevent progressive improvement.

Table 6: **Comprehensive cross-model evaluation demonstrating the model-specific nature of RSD.** Traces generated for one student (Transferred) fail to benefit other models and often degrade performance. However, when traces are generated specifically for a new student (Tailored), performance improves, confirming that distributional alignment must be unique to each model's architecture.

| Models | AIME24 | AIME25 | GPQA Diamond | MATH500 | Average |
|---|---|---|---|---|---|
| Qwen3-0.6B | 2.71 | 10.94 | 24.75 | 65.40 | 25.95 |
| + s1K-1.1 | 1.93 (-0.78) | 9.53 (-1.41) | 12.88 (-11.87) | 58.20 (-7.20) | 20.64 (-5.31) |
| + RSD-generated (Tailored) | 3.28 (+0.57) | 12.60 (+1.66) | 26.77 (+2.02) | 66.20 (+0.80) | 27.21 (+1.26) |
| Llama-3.2-1B-Instruct | 0.99 | 0.05 | 6.94 | 26.00 | 8.50 |
| + s1K-1.1 | 0.57 (-0.42) | 0.05 (0.00) | 9.47 (+2.53) | 10.00 (-16.00) | 5.02 (-3.48) |
| + RSD-generated (Transferred) | 0.42 (-0.57) | 0.05 (0.00) | 9.97 (+3.03) | 20.40 (-5.60) | 7.71 (-0.79) |
| + RSD-generated (Tailored) | 1.04 (+0.05) | 0.10 (+0.05) | 6.82 (-0.12) | 26.40 (+0.40) | 8.59 (+0.09) |
| Gemma-3-1B-IT | 0.73 | 0.52 | 3.60 | 41.00 | 11.46 |
| + s1K-1.1 | 0.00 (-0.73) | 0.00 (-0.52) | 2.53 (-1.07) | 13.20 (-27.80) | 3.93 (-7.53) |
| + RSD-generated (Transferred) | 0.10 (-0.63) | 0.00 (-0.52) | 3.72 (+0.12) | 17.00 (-24.00) | 5.21 (-6.25) |
| Phi-4-Mini | 2.66 | 1.41 | 16.79 | 53.80 | 18.66 |
| + s1K-1.1 | 5.52 (+2.86) | 3.80 (+2.39) | 16.48 (-0.31) | 51.20 (-2.60) | 19.25 (+0.59) |
| + RSD-generated (Transferred) | 5.89 (+3.23) | 4.22 (+2.81) | 18.62 (+1.83) | 57.20 (+3.40) | 21.48 (+2.82) |
| Phi-4-Mini-Reasoning | 24.90 | 21.15 | 44.13 | 73.60 | 40.95 |
| + s1K-1.1 | 20.94 (-3.96) | 19.84 (-1.31) | 28.54 (-15.59) | 79.60 (+6.00) | 37.23 (-3.72) |
| + RSD-generated (Transferred) | 15.00 (-9.90) | 17.34 (-3.81) | 26.20 (-17.93) | 75.00 (+1.40) | 33.39 (-7.56) |
| Qwen3-1.7B | 14.69 | 21.35 | 38.32 | 82.80 | 39.29 |
| + s1K-1.1 | 11.04 (-3.65) | 17.19 (-4.16) | 32.26 (-6.06) | 78.20 (-4.60) | 34.67 (-4.62) |
| + RSD-generated (Transferred) | 10.62 (-4.07) | 16.67 (-4.68) | 35.29 (-3.03) | 76.80 (-6.00) | 34.84 (-4.45) |
| + RSD-generated (Tailored) | 21.51 (+6.82) | 20.78 (-0.57) | 41.92 (+3.60) | 83.00 (+0.20) | 41.80 (+2.51) |
| Qwen3-4B | 20.05 | 20.52 | 45.08 | 86.80 | 43.11 |
| + s1K-1.1 | 22.76 (+2.71) | 26.88 (+6.36) | 43.56 (-1.52) | 86.60 (-0.20) | 44.95 (+1.84) |
| + RSD-generated (Transferred) | 17.60 (-2.45) | 22.55 (+2.03) | 43.12 (-1.96) | 84.80 (-2.00) | 42.02 (-1.09) |

Table 7: **Performance of iterative, multi-step RSD training.** Applying RSD in multiple cycles, where the student model is updated after each cycle, leads to performance degradation compared to a single training run.

| Models | AIME24 | AIME25 | GPQA Diamond | MATH500 | Average |
|---|---|---|---|---|---|
| Qwen3-0.6B | 2.71 | 10.94 | 24.75 | 65.40 | 25.95 |
| + RSD-generated ($p_{th}$=1%, single step, 15 epochs) | 3.28 | 12.60 | 26.77 | 66.20 | 27.21 |
| + RSD-generated ($p_{th}$=1%, three steps, 5 epochs each) | 1.93 | 9.06 | 22.22 | 61.60 | 23.70 |

# F  RSD TRACE LENGTHS ACROSS ARCHITECTURES

RSD's effectiveness is influenced by a student model's inherent linguistic style. Table 8 quantifies this by comparing the average token counts in traces generated for different student models. A contrast exists between the traces for Qwen3-0.6B, which average over 4,000 tokens, and those for Llama-3.2-1B-Instruct, which average only 1,081 tokens. The conciseness of the Llama-3.2-1B-Instruct traces, a reflection of its native style, provides an insufficient learning signal for complex reasoning, helping to explain the model's minimal performance gains when using this method.

Table 8: **Average token counts in RSD-generated traces across different student models.** Comparison shows dramatic differences between Qwen3-0.6B (with s1.1-7B teacher) and Llama-3.2-1B-Instruct (with DeepSeek-R1-Distill-Llama-8B teacher) across probability thresholds.

| Datasets | Average token count |
|---|---|
| RSD-generated ($p_{th}$=10%, tailored for Qwen3-0.6B) | 4156 |
| RSD-generated ($p_{th}$=3%, tailored for Qwen3-0.6B) | 4211 |
| RSD-generated ($p_{th}$=1%, tailored for Qwen3-0.6B) | 4266 |
| RSD-generated ($p_{th}$=0.3%, tailored for Qwen3-0.6B) | 4396 |
| RSD-generated ($p_{th}$=1%, tailored for Llama-3.2-1B-Instruct) | 1081 |

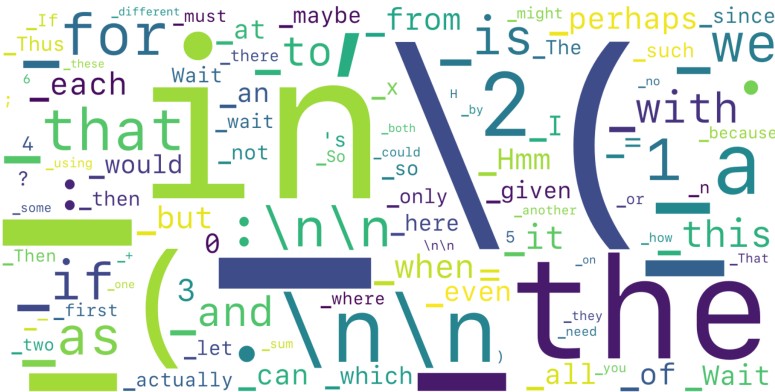

Figure 6: **Wordcloud of sub-1% probability tokens in s1K dataset traces.** Analysis reveals that problematic tokens frequently include logical connectors and transitional keywords that exceed the student model's distributional characteristics, validating RSD's threshold-based filtering approach.

## G    CHARACTERISTICS OF LOW PROBABILITY TOKENS

Figure 6 visualizes sub-1% probability tokens encountered in s1K traces under the Qwen3-0.6B student model. The analysis reveals that problematic tokens frequently include transition keywords representing reasoning patterns beyond the student model's current distributional characteristics.

## H    MMLU-PRO EVALUATION RESULTS

To validate the generalizability of RSD beyond mathematical and scientific reasoning, we evaluated performance on the MMLU-Pro benchmark. We specifically selected categories that demand logical deduction and reasoning: Law, Economics, Health, Psychology, Business, and Philosophy, totaling 4,849 question-answer pairs.

Table 9: **Performance on general domain reasoning (MMLU-Pro).** We report accuracy on selected categories requiring logical deduction (Law, Economics, Health, Psychology, Business, and Philosophy). Consistent with mathematical benchmarks, direct distillation degrades performance while RSD yields improvements over the base model.

| Models | MMLU-Pro (Selected) |
| --- | --- |
| Qwen3-0.6B | 30.93 |
| + s1K-1.1 | 30.54 |
| + RSD-generated ($p_{th}$=1%) | 31.39 |

As shown in Table 9, the trend observed in math and science benchmarks holds for general domain reasoning. Direct distillation of s1K-1.1 traces degrades performance relative to the base model, while the model trained on RSD-generated traces achieves improved accuracy, confirming that distributional alignment benefits extends to general reasoning.

## I    DETAILED S1K-1.1 VS. RSD TRACE COMPARISONS

The following comparisons between original s1K-1.1 traces and their RSD counterparts demonstrate how RSD systematically filters high-surprisal tokens while preserving the logical structure and reasoning complexity of the original traces across diverse problem types and reasoning patterns.

918
919
920
921
922
923
924
925
926
927
928
929
930
931
932
933
934
935
936
937
938
939
940
941
942
943
944
945
946
947
948
949
950
951
952
953
954
955
956
957
958
959
960
961
962
963
964
965
966
967

<|im_start|>system\nYou are Qwen, created by Alibaba Cloud. You are a helpful assistant.<|im_end|>\n<|im_start|>user\nGiven a rational number, write it as a fraction in lowest terms and calculate the product of the resulting numerator and denominator. For how many rational numbers between 0 and 1 will $ 20_{}^{}!$ be the resulting product?<|im_end|>\n<|im_start|>assistant\n<think>\nAlright, so I need to figure out how many rational numbers between 0 and 1 have a product of the numerator and denominator (when written in lowest terms) equal to 20 factorial. Let me start by understanding the problem.\n\nFirst, a rational number between 0 and 1 can be written as a fraction \\(\\frac{a}{b}\\) where \\(0 < a < b\\) and \\(a\\) and \\(b\\) are coprime positive integers. The product \\(a \times b\\) is said to be equal to 20! which is a huge number. So I need to find all pairs \\((a, b)\\) such that \\(a \times b = 20!\\), \\(0 < a < b\\), and \\(\\gcd(a, b) = 1\\). Then count how many such pairs exist.\n\nLet me break down the problem.\n\nGiven that \\(a \times b = 20!\\) and \\(\\gcd(a, b) = 1\\), it means that \\(a\\) and \\(b\\) are coprime [...] \\). Since \\(a\\) and \\(b\\) are coprime, each prime factor in 20! must go entirely to \\(a\\) or entirely to \\(b\\). Hence, each coprime pair corresponds to a subset of the prime factors of 20!, where \\(a\\) is the product of the primes in the subset and \\(b\\) is the product of the remaining primes. However, since 20! is not just the product of primes, but the product of primes raised to various powers. Wait, so maybe I need to rethink that.\n\nWait, actually, 20! has prime factors with exponents. For example, 2^18, 3^8, 5^4, etc., up to the primes less than or equal to 20. But if I need two coprime numbers \\(a\\) and \\(b\\) such that \\(a \times b = 20!\\), then each of the prime powers in the factorization of 20! must go entirely to one of the numbers \\(a\\) or \\(b\\). So for each prime factor, we assign it to either \\(a\\) or \\(b\\), but not both.\n\nTherefore, the number of such coprime pairs is \\(2^k\\), where \\(k\\) is the [...] the assignment where a = 1 (all primes assigned to b) and b = 20! (there's exactly one such case), and a can be 1, then proceed. Wait, 1 is the product of assigning no primes to a (all primes to b), so that's included.\n\nBut if a and b can be written as a = 1 and b = 20!, yes, that's a coprime pair, likewise a=2^18 and b= (20!)/2^18 which is still an integer. Similarly for all subsets. Wait, but when we write a and b as coprime factors, the way I'm thinking is that the assignment is of each prime as a binary choice (to a or to b). So 2^8 equals 256 possible assignments, and each assignment defines a unique coprime pair (a,b) where a * b=20!. Since each assignment is unique, each ordered pair is distinct, even if a and b are swapped.\n\nSince such ordered pairs can have a < b or a > b, so then to get the \\((a,b) with a < b, we divide by 2, which gives 128.\n\nBut wait, can it be? Let me check with a smaller factorial.\n\n[...]

0 and 1 such that when written in lowest terms as a/b, the product of numerator and denominator is 20!=1×2×3...×20 is given by 2^{number of prime factors of 20! -1} =2^{8-1}=128.\n\n**Final Answer**\n\boxed{128}\n</think>\n\nAnswer: Given a rational number between 0 and 1, we need to write it as a fraction in lowest terms \\(\\frac{a}{b}\\) and calculate the product of the resulting numerator and denominator. We need to determine how many such rational numbers exist where the product \\(a \times b = 20!\\).\n\n1. **Prime Factorization of 20!**:\n  - The prime factors of 20! are 2, 3, 5, 7, 11, 13, 17, and 19. These are 8 distinct primes.\n\n2. **Coprime Pairs**:\n  - For \\(a \times b = 20!\\) and \\(\\gcd(a, b) = 1\\), each prime factor of 20! must be assigned entirely to either \\(a\\) or \\(b\\). This results in \\(2^8 = 256\\) coprime pairs \\((a, b)\\).\n\n3. **Counting Pairs \\(a < b\\)**:\n  - Since \\(a\\) and \\(b\\) are coprime and \\(a \times b = 20!\\), each pair \\((a, b)\\) and \\((b, a)\\) are distinct. To find pairs where \\(a < b\\), we divide the total number of pairs by 2, excluding the case \\(a = b\\) (which is impossible since 20! is not a square).\n\n4. **Result**:\n  - The number of coprime pairs \\((a, b)\\) with \\(a < b\\) is \\(\\frac{256}{2} = 128\\).\n\nThus, the number of rational numbers between 0 and 1 for which the product of the numerator and denominator in lowest terms is \\(20!\\) is \\(\\boxed{128}\\).<|im_end|>\n

<|im_start|>system\nYou are Qwen, created by Alibaba Cloud. You are a helpful assistant.\nYou must respond to every query in the following manner:\nFirst, provide a step-by-step logical exploration of the problem.\nThen, provide a clear and direct response based on your reasoning, with the final answer enclosed in \boxed{}.<|im_end|>\n<|im_start|>user\nGiven a rational number, write it as a fraction in lowest terms and calculate the product of the resulting numerator and denominator. For how many rational numbers between 0 and 1 will $ 20_{}^{}!$ be the resulting product?<|im_end|>\n<|im_start|>assistant\n<think>\nOkay, let me try to work through this problem step by step. So the question is: Given a rational number between 0 and 1, write it as a fraction in lowest terms and calculate the product of the numerator and denominator. How many such rational numbers will result in the product 20!?\n\nFirst, I need to understand what a rational number between [...] means their greatest common divisor (gcd) is 1.\n\nThe problem says we need to calculate the product of the numerator and denominator (i.e., p * q) for such fractions, and we need to find how many such fractions have p * q equal to 20!.\n\nSo, first, I need to figure out all the fractions p/q where 0 < p/q < 1, p and q are coprime, and p * q = 20!.\n\nWait, hold on. If p * q = 20! and p and q are coprime, then p and q must be divisors of 20! and coprime to each other. Since they are coprime, they have no common prime factors. But since their product is 20!, that means p and q are two coprime integers whose product is 20!. So essentially, we need to find all pairs of coprime positive integers (p, q) such that p * q = 20! and p/q is between 0 and 1. Therefore, q must be greater than p, because p/q < 1.\n\nSo the problem reduces to finding all coprime pairs (p, q) with p * q = 20! and q > p.\n\n[...]

Wait, when we assign each prime to either p or q, since the primes in the factorization of 20! include all primes up to 19. Therefore, the number of coprime pairs (p, q) with p < q is equal to half the total number of coprime pairs, right? Because for each pair (p, q), either p < q or p > q. Since we are considering only the ones where p < q, which would be half of the total, but we need to check if the pairs are ordered. However, when considering ordered pairs, the number of pairs where p < q is equal to the number of pairs where p > q, except when p = q. But here, p and q are coprime, so p cannot equal q unless p = q =1, but p*q =20! implies that p and q are factors of 20!, so the only way p*q =1 would be if p=q=1, but 1 is not a prime factor of 20!. Therefore, p and q must be distinct.\n\nTherefore, the number of coprime pairs (p, q) with p < q is equal to the total number of coprime pairs divided by 2. However, since when p [...] distinct primes, which is 8 for 20!), then the answer is 2^{k}/2=2^{k-1}.\n So 2^{8-1}=128. So, 128 is the answer. Therefore, the answer is \boxed{128}.\n\n**Final Answer**\n\boxed{128}\n</think>\n\nTo determine the number of rational numbers between 0 and 1, written in lowest terms as a fraction $ \frac{p}{q} $, such that the product of the numerator and denominator equals 20!, we start by understanding the problem constraints:\n\n1. A rational number between 0 and 1 can be expressed as a fraction $ \frac{p}{q} $ where $ p $ and $ q $ are coprime (i.e., gcd(p, q) = 1) and $ 0 < \frac{p}{q} < 1 $.\n2. For such fractions, we need to compute the product $ p \cdot q $ and count how many such products equal 20!.\n\n### Key Insight:\n- Since $ p $ and $ q $ must be coprime, each prime factor in 20! must be assigned to either $ p $ or $ q $. This leads to $ 2^k $ coprime pairs (p, q), where $ k $ is the number of distinct prime factors of 20!.\n\n### Prime Factorization of 20!:\nThe prime factors of 20! are 2, 3, 5, 7, 11, 13, 17, and 19. Thus, there are 8 distinct prime factors.\n\n### Calculation:\n- The number of coprime pairs (p, q) is $ 2^8 = 256 $ (including ordered pairs where p or q could be 1).\n- Since we are interested in unordered pairs where $ p < q $, we divide by 2, resulting in $ \frac{256}{2} = 128 $ such pairs.\n\nThus, the number of rational numbers between 0 and 1 where the product of the numerator and denominator equals 20! is:\n\n$$\n\boxed{128}\n$$ <|im_end|>

968
969
970
971

Figure 7: **Detailed trace comparison reveals RSD's distributional alignment.** Selectively presented sections show logically similar points from s1K-1.1 traces (left) with numerous sub-1% probability tokens (red highlights) and RSD-generated traces (right) with smooth probability transitions.

