# OpenReview forum: "In Their Own Words: Reasoning Traces Tailored for Small Models Make Them Better Reasoners"
_ICLR.cc/2026/Conference — Submitted to ICLR 2026_

### Official Review · Reviewer_HoV7 · 2025-10-17

**Soundness:** 2
**Presentation:** 3
**Contribution:** 2
**Rating:** 2
**Confidence:** 4

**Summary:**

This paper proposes a simple method to prevent performance degradation when distilling reasoning capabilities from larger teacher models to smaller student models. The simple method is to set a threshold to filter out the tokens in the reasoning traces with low probabilities and replace them with the student's generation.

While the problem is well-motivated and the idea is interesting, I have two major concerns regarding the methodological robustness and the novelty of the contribution.

**Strengths:**

* The problem is well-motivated
* The paper is well-written and easy to understand

**Weaknesses:**

* The proposed method lacks sufficient empirical support and appears ad-hoc. This is seen from 2 perspectives: (1) The method's effectiveness is demonstrated on only a single dataset (s1K). This narrow evaluation makes it difficult to assess the generalizability of the findings. The threshold (0.01) seems to be a hyper-parameter that could be overfitted to this specific dataset's characteristics. (2) As the authors report in Sec 5.4, iteratively applying RSD leads to perf degradation, this suggests that RSD is not fundamentally improving the student's reasoning capability (i.e., I would treat the student at iter K as a new student I have never met and expect it it be improved).
* The novelty is unclear in the context of prior work. The core idea of aligning a teacher's output with a student's distribution to create more effective training data has been explored in "Reinforcement Learning Teachers of Test Time Scaling" (Cetin et al., 2025). In that work, the teacher is trained with a reward function that includes a KL term to explicitly minimize the difference between the teacher's and the student's distributions over the generated explanation tokens. This is fundamentally the same goal as RSD.

**Questions:**

1. Can the authors test their method on more datasets?
2. How is your method related to RLT (see weakness \#2)?

---

> ### Author Response · Authors · 2025-12-01
>
> Thank you for your review. We appreciate your recognition of the problem's motivation. We address your concerns regarding empirical support and novelty below.
>
> ---
>
> ## Generalizability across datasets (W1, Q1)
>
> We acknowledge the concern regarding single-dataset evaluation. To address this, **we conducted additional experiments using OpenThoughts3-1.2M**. We filtered the dataset for math and science domains, removed truncated traces, deduplicated QA pairs, and randomly sampled 1,000 examples to match the s1K experimental setup. The results mirror our findings on s1K:
>
> | Model | AIME24 | AIME25 | GPQA Diamond | MATH500 | Average |
> | :--- | :--- | :--- | :--- | :--- | :--- |
> | Qwen3-0.6B (Base) | 2.71 | 10.94 | 24.75 | 65.40 | 25.95 |
> | + OpenThought s3 1k subset (Direct Distill) | 1.82 | 6.35 | 22.41 | 54.00 | 21.15 |
> | + OpenThought s3 1k subset RSD p_th=1% | 4.01 | 9.58 | 22.92 | 64.40 | 25.23 |
>
> While RSD did not surpass the base model's average in this instance, it is important to highlight that on AIME24, the most difficult reasoning benchmark, RSD achieved the best performance (4.01), outperforming the base model (2.71). **This confirms that RSD's filtering mechanism is robust across datasets, preventing the catastrophic degradation seen in direct distillation and enabling gains in complex reasoning tasks.**
>
> ---
>
> ## Iterative Training and Methodological Robustness (W1)
>
> **The degradation during iterative training (Section 5.4) should be viewed in the context of the broader, unsolved research challenge of Model Collapse [1].** When a model is recursively trained on its own output distribution, even with teacher guidance, the distribution tends to narrow, reducing diversity and capability. This is a general characteristic of recursive learning paradigms rather than a specific flaw in the RSD mechanism itself. RSD successfully serves its primary purpose: bridging the distributional gap between a strong teacher and a weak student to generate effective training data.
>
> ---
>
> ## Novelty and Relation to RLT (W2, Q2)
>
> While RSD and RLT (Cetin et al., 2025) share the high-level goal of aligning teacher and student distributions, the approaches are fundamentally different in implementation and cost:
> - RLT (Teacher-Training): **Requires training the large teacher model using RL with a KL-divergence constraint towards the student.** This is computationally expensive and requires modifying teacher weights.
> - RSD (Data Generation): **Is a data synthesis mechanism. It does not require training the teacher.** Instead, it utilizes the teacher "as-is" to generate aligned training data for the student by filtering tokens during the generation process.
>
> This distinction makes RSD significantly **more accessible for researchers with limited compute who wish to distill open-weights models (like DeepSeek-R1 or Qwen) into smaller architectures without the heavy cost of retraining the teacher model**. Beyond immediate improvements in distillation, this work furthers our understanding of the probability distribution characteristics intrinsic to next-token prediction machines, demonstrating that the capacity for reasoning is inextricably interlinked with the model's distributional landscape. We will clarify these distinctions and insights in the final paper.
>
> ---
>
> [1] Shumailov, I. et al. AI models collapse when trained on recursively generated data. Nature 631, 755–759 (2024)

---

### Official Review · Reviewer_QJZQ · 2025-10-25

**Soundness:** 1
**Presentation:** 3
**Contribution:** 2
**Rating:** 2
**Confidence:** 3

**Summary:**

This paper argues that strong reasoning traces from large "teacher" models can be suboptimal for LLM reasoning distillation in case the distributional misalignment between teacher and student is too large. Thus, the authors propose "RSD," a method to collect reasoning traces from large models by generating tokens and either accepting or rejecting them based on the student's probability, essentially guiding the teacher closer to the student distribution. The authors evaluate on AIME, GPQA, and MATH, showing that while distilling with the s1 dataset from [1] can hurt the average performance of reasoning/post-trained models, RSD can provide some improvements.

[1] Muennighoff, Niklas, et al. "s1: Simple test-time scaling." arXiv preprint arXiv:2501.19393 (2025).

**Strengths:**

- The concept that using reasoning traces that the student LLM struggles to understand is suboptimal is something that is intuitively logical and is consistent with prior results.
- I found the presentation level of the paper to be high, with the methodology and results being clearly presented.
- The authors share their anonymized code with the submission, which is great for research transparency and reproducibility.

**Weaknesses:**

**Major**
My main concerns are centered around the evaluation performed and the results:
1) Looking at the shared code, for all main experiments, the authors consider already post-trained versions of new models, with a focus on the Qwen3 family. This version of Qwen3 was already distilled with a much broader and higher quality closed-source of diverse reasoning data from the recent Qwen3 paper. In contrast, the open source S1 dataset is older and not really considered competitive even when compared to other open-source efforts such as [1]. To this end, I do not think it is surprising that the Qwen3-06B model's performance deteriorates. Thus, I am unsure how this provides any validation for the proposed hypothesis that low-probability tokens and not just "lower-quality" tokens are the cause of the observed phenomena. To really test the introduced hypothesis, I think the focus of the experiments should be models that have not already been subjected to large-scale reasoning post-training (e.g., Qwen2.5 instruct and/or Qwen3 base) together with broader and more competitive reasoning datasets (such as OpenThoughts [1]). While currently, the authors have done some preliminary investigations in Figure 4, I do not think that just varying the model axis with other competitive and/or reasoning models is enough.
2) While the authors claim the hyperparameters are "following the s1 training recipe" [ln 243], their hyperparameters actually differ. For instance, the models from the original s1 paper seem to train for 5 epochs, while the authors of this paper report training for 15, potentially artificially increasing the gap caused by training on this relatively suboptimal, older dataset. I would appreciate it if the authors could explain this divergent choice, or whether I am perhaps misreading the s1 hyperparameters.
3) From Table 1, the best reported performance of the method after sweeping the percentile hyperparameter is only increased by a very small percentage on evaluations with very few problems (e.g., less than a single average question of improvement on AIME24 and AIME25). Other variations in hyperparameters seem to even hurt performance. These results do not seem statistically significant and suggest a considerable brittleness to hyperparameters. Moreover, the performances reported for Qwen3B-0.6 and all its post-training variations are visibly lower than what was attained in the Qwen3 paper (2.7 vs 10.7 for AIME24, 10.9 vs 15.7 for AIME25, 24.8 vs 27.9 for GPQA, 65.4 vs 77.6 for MATH500). I would encourage the authors to provide an explanation for this gap.


**Other**
1) The idea that models can benefit from their own current understanding, guided by teachers, and that teachers themselves can benefit from going towards the student's distribution is something very related to the StaR family of methods and other recent work on reasoning teachers [2, 3, 4, 5]. I think these are all quite relevant related works whose connection to RSD should be explained in the paper.

[1] https://huggingface.co/datasets/open-thoughts/OpenThoughts3-1.2M

[2] Zelikman, Eric, et al. "Star: Bootstrapping reasoning with reasoning." Advances in Neural Information Processing Systems 35 (2022): 15476-15488.

[3] Li, Xiaochuan, Zichun Yu, and Chenyan Xiong. "Montessori-instruct: Generate influential training data tailored for student learning." arXiv preprint arXiv:2410.14208 (2024).

[4] Guan, Xinyu, et al. "rStar-Math: Small LLMs Can Master Math Reasoning with Self-Evolved Deep Thinking." arXiv preprint arXiv:2501.04519 (2025).

[5] Cetin, Edoardo, Tianyu Zhao, and Yujin Tang. "Reinforcement Learning Teachers of Test Time Scaling." arXiv preprint arXiv:2506.08388 (2025).

**Questions:**

Overall, I think the direction of providing better reasoning traces with distillation through a new decoding technique can be very valuable, and I appreciate the clear exposition the paper provided. However, I found the evaluation to be quite rushed, with empirical flaws and inconsistencies that I do not think provide concrete evidence of the proposed method's effectiveness. For these reasons, I do not think the paper is ready for acceptance at this stage.

---

> ### Author Response · Authors · 2025-12-01
>
> We appreciate your detailed review and the references to related work. We would like to address your concerns regarding the experimental setup and datasets.
>
> ---
>
> ## Dataset Quality and Validity of the Hypothesis (W1)
>
> We respectfully disagree that the performance degradation is solely due to s1K being "older" or "lower quality" than Qwen3's training data. **If the dataset quality were the only issue, both Direct SFT and RSD should fail.** However, RSD achieves a 4.9% improvement over Direct SFT using the exact same question-answer pairs. This strongly suggests that distributional misalignment is the primary cause of the failure, rather than the quality of the questions in the s1K dataset.
>
> To further validate this, **we conducted a new experiment using OpenThoughts3-1.2M** (the high-quality dataset you mentioned). We processed the dataset by selecting only math and science domains, filtering out truncated traces that exceeded the context limit, and deduplicating to ensure unique Question-Answer pairs. From the resulting pool of approximately 35k valid traces, we randomly sampled 1,000 examples to ensure a fair, size-controlled comparison with our s1K baseline.
>
> | Model | AIME24 | AIME25 | GPQA Diamond | MATH500 | Average |
> | :--- | :--- | :--- | :--- | :--- | :--- |
> | Qwen3-0.6B (Base) | 2.71 | 10.94 | 24.75 | 65.40 | 25.95 |
> | + OpenThought s3 1k subset (Direct Distill) | 1.82 | 6.35 | 22.41 | 54.00 | 21.15 |
> | + OpenThought s3 1k subset RSD p_th=1% | 4.01 | 9.58 | 22.92 | 64.40 | 25.23 |
>
> The results reinforce our core hypothesis: **Direct distillation causes severe degradation** (dropping from 25.95 to 21.15), confirming that even with state-of-the-art data, distributional misalignment remains a critical barrier. While RSD on this specific sample did not exceed the base model's average score, it successfully mitigated the degradation caused by direct distillation. **Most notably, on AIME24—the most challenging benchmark requiring the deepest reasoning capabilities—RSD achieved the highest performance** (4.01), significantly outperforming both the base model (2.71) and direct distillation (1.82). This suggests that RSD is particularly effective at preserving the complex reasoning chains required for hard problems.
>
> ---
>
> ## Use of Post-trained Models (W1)
>
> Our research question specifically investigates the transfer failure in small models that are already capable but struggle with complex teacher traces. Using post-trained models allows us to isolate the distillation dynamics without the noise of teaching the model basic instruction-following from scratch. **The fact that we can improve a model that has already undergone extensive post-training further highlights the efficacy of RSD.**
>
> ---
>
> ## Hyperparameters and Training Details (W2, W3)
>
> - Epochs: We performed a hyperparameter sweep for the baseline (Direct SFT) and found that **15 epochs yielded the best performance for this specific dataset size** (1k samples). We applied the same setting to RSD to ensure a fair comparison where the baseline was strongest.
> - Performance Gap: The performance gap between our reported numbers and the Qwen3 technical report **stems largely from evaluation context limits**. We used 8k context due to compute constraints, whereas the technical report utilized 32k. While the absolute numbers differ, the relative ranking between methods remains a valid signal of RSD's effectiveness.
>
> ---
>
> ## Related Work (Other)
>
> We agree that the STaR family and recent works like RLT are highly relevant. We will expand our related work section to explicitly position RSD against these methods. Specifically regarding RLT, while we share the high-level goal of aligning teacher and student distributions, our approaches differ fundamentally in implementation and computational cost. **RLT achieves alignment by training the large teacher model using RL with a KL-divergence constraint** towards the student, a process that requires modifying teacher weights and significant computational resources. **In contrast, RSD is a data synthesis mechanism that operates at inference time, utilizing the teacher "as-is"** to generate aligned training data by filtering tokens during the generation process. This distinction renders RSD significantly more accessible for researchers with limited compute who wish to distill open-weights models into smaller architectures without the heavy cost of retraining the teacher.
>
> Beyond immediate improvements in distillation, this work furthers our understanding of the probability distribution characteristics intrinsic to next-token prediction machines, demonstrating that the capacity for reasoning is inextricably interlinked with the model's distributional landscape. We will clarify these distinctions and insights in the final paper.

---

### Official Review · Reviewer_w4jo · 2025-10-31

**Soundness:** 2
**Presentation:** 2
**Contribution:** 2
**Rating:** 2
**Confidence:** 3

**Summary:**

This paper addresses the counterintuitive failure of transferring reasoning capabilities from large to small language models (LLMs/SLMs) via supervised fine-tuning (SFT). The authors attribute this degradation to "distributional misalignment," where teacher-generated reasoning traces contain tokens with extremely low probability under the student model's distribution, creating learning barriers. To resolve this, they propose Reverse Speculative Decoding (RSD), a novel mechanism where the student model dictates token acceptance based on its own probability threshold, generating model-specific, distributionally-aligned reasoning traces. Experiments with Qwen3-0.6B demonstrate that while direct distillation significantly degrades performance, training on RSD-generated traces yields a measurable performance gain.

**Strengths:**

（1）The paper provides a deep analysis of the counterintuitive failure in distilling reasoning capabilities to SLMs, identifying and verifying "distributional misalignment" as the core reason for performance degradation in models like Qwen3-0.6B.
（2）The proposed Reverse Speculative Decoding (RSD) method is conceptually sound and highly intuitive. By filtering out low-probability tokens from the teacher's reasoning trace before fine-tuning, the method successfully delivers a measurable performance boost.

**Weaknesses:**

（1）While $p_{th}=1\%$ is determined as the optimal value, the paper would benefit from a deeper analysis of how different $p_{th}$ values impact the semantic quality and reasoning logic correctness of the traces.
（2）Appendix Table 4 suggests that effective distillation requires tailored RSD traces for different student models. However, some models, like Qwen3-4B, lack customized experiments, leaving ambiguity about whether RSD is universally effective across all model architectures.

**Questions:**

（1）The paper indicates a need to customize RSD traces for different models. Could the authors confirm if performance gain is guaranteed for all models with tailored customization, or are certain architectures fundamentally unsuitable? Analysis of any failure cases would lead to deeper experimental conclusions.
（2）In Appendix Table 4, why does direct distillation of the teacher's reasoning trace yield a performance gain for the Qwen3-4B model without failure, unlike the 0.6B model? Is this related to the student model's size, and how does this observation align with the core "distributional misalignment" hypothesis?
（3）When the student model rejects the teacher's low-probability token, will sampling directly from the student's distribution potentially compromise the logical strictness or soundness of the teacher's intended reasoning path?

---

> ### Author Response · Authors · 2025-12-01
>
> Thank you for your constructive review and for highlighting the intuition behind RSD. We are happy to clarify the mechanism and scope of our method.
>
> ---
>
> ## Qualitative analysis of RSD traces & Impact on logic (W1, Q3)
>
> To investigate whether RSD's filtering affects the quality of reasoning, we conducted a pairwise comparison using GPT-5-nano as a judge. We sampled trace pairs (original s1K-1.1 vs. RSD p_th=1% and asked the evaluator to compare them on "Logical Correctness" and "Logical Tractability".
>
> | Metric | s1K-1.1 | RSD-generated (p_th=1%) |
> | :--- | :--- | :--- |
> | Logical Correctness | 53.89% | 46.11% |
> | Logical Tractability | 51.67% | 48.33% |
>
> The results show that RSD traces exhibit a slight degradation in both logical correctness and tractability compared to the original teacher traces. This answers the question regarding whether sampling from the student compromises logic: Yes, there is a trade-off. **When the student rejects a teacher token and samples from its own distribution, it risks choosing a logically incorrect path.** This is further evidenced by the drop in "Correctly solved" problems during generation (from 1000 s1K-1.1 traces generated with DeepSeek-R1 to 180 in RSD, as shown in Table 2).
>
> However, this trade-off is central to the RSD mechanism. **Our experiments show that training on these fewer but aligned traces leads to a better final model (Table 1).** The student learns more effectively from slightly imperfect but understandable traces than from perfect traces it cannot comprehend.
>
> ---
>
> ## Model suitability and Qwen3-4B (W2, Q1, Q2)
>
> You correctly noted that Qwen3-4B improves with direct distillation, unlike the 0.6B model. This supports our "Distributional Misalignment" hypothesis. The 4B model has sufficient internal representation capacity to process the teacher's (s1K-1.1) logic directly. **RSD specifically targets the students with limited representational capacity (e.g., transferring to <4B models) where direct distillation fails due to capacity constraints.**
>
> Regarding suitability: RSD requires the student to be "reasoning-enabled" (i.e., instruction-tuned to lengthy reasoning formats). **As seen with the case of Llama-3.2-1B-Instruct, if the model is not primed for reasoning, it may reject valid reasoning steps simply due to format unfamiliarity.** Thus, RSD guarantees gains for reasoning-enabled small models where capacity is the bottleneck, but not necessarily for models with fundamentally different behavioral training.

---

### Official Review · Reviewer_AcoZ · 2025-11-03

**Soundness:** 3
**Presentation:** 3
**Contribution:** 3
**Rating:** 6
**Confidence:** 2

**Summary:**

This paper addresses the counterintuitive phenomenon where small language models (SLMs) experience performance degradation when fine-tuned on high-quality reasoning traces from larger teacher models. The authors identify that this failure stems from distributional misalignment: teacher traces contain low-probability tokens that exceed the student's internal representation capacity. They propose Reverse Speculative Decoding (RSD), a mechanism that inverts traditional speculative decoding by having the teacher propose tokens while the student determines acceptance based on its own probability distribution (rejecting tokens below a threshold p_th).

**Strengths:**

- Novel inversion of speculative decoding paradigm (teacher-proposed, student-approved vs. standard student-proposed, teacher-approved)
- Comprehensive cross-model evaluation testing transferability both within-family (Qwen3 0.6B→1.7B, 4B) and inter-family (to LLaMA, Gemma, Phi)
- Well-motivated problem with clear empirical evidence (20.5% degradation from s1K-1.1)

**Weaknesses:**

- All experiments limited to mathematical reasoning only
- No systematic exploration of teacher model selection (different architectures, training methods, capabilities)
- Missing systematic analysis of which model properties (size, training data, architecture) predict RSD effectiveness

**Questions:**

- Is there a theoretical framework predicting which student-teacher pairs will benefit from RSD?
- What if you use the student's own successful rollouts as "teacher" traces?

---

> ### Author Response · Authors · 2025-12-01
>
> We appreciate your thoughtful feedback and recognition of the novelty of our reverse speculative decoding paradigm. We address your questions and concerns below.
>
> ---
>
> ## Theoretical framework for student-teacher pairs (W2, W3, Q1)
>
> We posit that **successful RSD application relies on two key constraints** regarding the student model:
>
> 1. **The "Reasoning-Enabled" Constraint**: RSD cannot instill reasoning behavioral patterns like extensive step-by-step deduction into a model from scratch. The student model must be previously exposed to reasoning patterns (e.g., via instruction tuning). As seen in our Llama-3.2-1B experiments (Section 5.2), if a student is not sufficiently primed for reasoning, it tends to truncate the trace prematurely because the teacher's lengthy reasoning falls too far outside the student’s behavior distribution.
> 2. **The "Capacity Gap" Constraint**: RSD is designed specifically for scenarios where the representational capacity of a student is limited enough to cause distributional misalignment from teacher traces. For models with larger representational capacities (like Qwen3-4B, as discussed in Appendix D), direct distillation works effectively. We target the sub-4B regime (e.g., 0.6B) where direct distillation counterintuitively fails.
>
> ---
>
> ## Generalization beyond mathematical reasoning (W1)
>
> To address the concern that our evaluation was limited to math/science, **we conducted additional experiments on MMLU-Pro**, specifically selecting categories requiring logical deduction: Law, Economics, Health, Psychology, Business, and Philosophy (Totaling 4,849 QA pairs).
>
> As shown in the table below, **the trend observed in math benchmarks holds for general domain reasoning**. Direct distillation of s1K-1.1 traces degrades performance compared to the base model, while RSD-generated traces improve upon the baseline.
>
> | Models | MMLU-Pro (Selected Categories) | AIME24 | AIME25 | GPQA Diamond | MATH500 |
> | :--- | :--- | :--- | :--- | :--- | :--- |
> | Qwen3-0.6B (Base) | 30.93 | 2.71 | 10.94 | 24.75 | 65.40 |
> | + s1K-1.1 (Direct Distill) | 30.54 | 1.93 | 9.53 | 12.88 | 58.20 |
> | + RSD-generated (p_th=1%) | 31.39 | 3.28 | 12.60 | 26.77 | 66.20 |
>
> ---
>
> ## Student’s own successful rollouts as teacher traces (Q2)
>
> **We investigated this exact scenario in our "Self-distill" baseline (Table 1)**, where the student generates its own traces without teacher guidance. This **resulted in no performance gain** (Avg 25.80 vs 25.95). Furthermore, in Section 5.6, we also conducted a compute-equivalent experiment where the self-distillation budget matched RSD's rejection sampling budget (203 attempts). **Even with increased compute, self-distillation failed to outperform the base model.** This confirms that teacher guidance—filtered through RSD—is essential for improvement; the student cannot simply bootstrap itself without the superior reasoning patterns provided by the teacher.

---

### Author Response · Authors · 2025-12-02
**Rebuttal Summary Comment**

We are sincerely grateful to all reviewers for their constructive feedback. We have updated our paper to incorporate the requested experiments and analyses (modifications marked in blue in the revised PDF). Below, we summarize how we addressed the key concerns regarding dataset generalizability, model applicability, and novelty.

---

## **1) Validation across Diverse Datasets and Domains**
Reviewers **(QJZQ, HoV7, AcoZ)** raised concerns that our findings were limited to a single dataset (s1K) or domain (math). We have addressed this through two new extensive experiments:
*   **New Dataset (OpenThoughts3-1.2M):** To refute the claim that performance degradation was due to s1K data quality, we ran experiments on the recent OpenThoughts3 dataset. Results confirm our core hypothesis: Direct distillation still causes degradation (dropping from 25.95% to 21.15%), while RSD recovers performance (25.23%). Notably, **on AIME24, the most challenging benchmark, RSD achieved the highest performance** (4.01%), outperforming both the base model (2.71%) and direct distillation (1.82%).
*   **New Domain (MMLU-Pro):** We extended our evaluation to general reasoning categories (Law, Psychology, Philosophy, etc.). RSD consistently improved over the baseline (31.39% vs 30.93%), whereas direct distillation degraded performance (30.54%), proving RSD generalizes beyond mathematics.

---

## **2) Conceptual Distinction from Related Work (RLT)**
Reviewers **(QJZQ, HoV7)** asked for clarification regarding the relationship between RSD and Reinforcement Learning Teachers (RLT).
*   We clarified that while both methods aim to align distributions, **RLT requires computationally expensive training of the teacher model** using RL.
*   In contrast, **RSD is a data synthesis mechanism that operates at inference time**, allowing researchers to utilize open-weights teachers "as-is" without modification. This makes reasoning distillation accessible for settings where retraining a teacher model is infeasible.

---

## **3) Trade-offs in Logical Correctness vs. Learnability**
Reviewer **(w4jo)** asked if RSD filters compromise the logical strictness of traces.
*   We conducted a **GPT-5-based qualitative evaluation**, which revealed that RSD traces indeed exhibit a slight degradation in logical correctness compared to raw teacher traces.
*   However, we argue—and our training results confirm—that this is a necessary trade-off. **The student model learns significantly better from slightly imperfect but distributionally aligned traces than from "perfect" teacher traces that exceed its representational capacity.**

---

## **4) Constraints on Student Model Applicability**
Reviewers **(w4jo, AcoZ)** asked for a theoretical framework regarding which models benefit from RSD.
*   We clarified the **"Capacity Gap"** constraint: Larger students (e.g., Qwen3-4B) have sufficient capacity to handle teacher traces directly, making RSD unnecessary (as confirmed by our experiments). RSD is specifically critical for the sub-4B regime (e.g., 0.6B) where capacity constraints cause standard distillation to fail.
*   We also established the **"Reasoning-Enabled"** constraint, explaining why models not primed for extensive reasoning (like Llama-3.2-Instruct) struggle to utilize RSD effectively.

---

## **5) Compute-Equivalent Baselines**
Reviewer **(AcoZ)** queried if student-only generation could achieve similar results.
*   We highlighted our compute-equivalent experiment (section 5.6 in the original version, which moved to section 5.9 in the revised version), showing that even when given the same rejection sampling budget (203 attempts) as RSD, **self-distillation failed to outperform the base model**. This confirms that teacher guidance—specifically filtered through RSD—is essential.

---

We believe these additional experiments and clarifications firmly establish RSD as a vital technique for enabling reasoning transfer to compact models, solving a distillation problem that standard SFT cannot address.

---

### Meta-Review · Area_Chair_5HSd · 2025-12-21

**Summary:**

It is difficult to promote a paper with three non-positive reviews out of 4.

**Reviewer Concerns:**

Reviewers' concerns are partially addressed, but it is impossible to promote the paper.

**Reviewer Scores:**

Possibly, the level could not be reverted to accept.

---

### Decision · Program_Chairs · 2026-01-26

Reject